# Comparative Study on Different Interpolation Methods and Source Analysis of Soil Toxic Element Pollution in Cangxi County, Guangyuan City, China

Jiajun Zhang [1], Junsheng Peng [1,*], Xingyi Chen [1], Xinyi Shi [1], Ziwei Feng [1], Yichen Meng [1], Wende Chen [1,*] and Yingping Liu [2]

1    College of Geography and Planning, Chengdu University of Technology, Chengdu 610059, China; zhangjiajun@stu.cdut.edu.cn (J.Z.); 202212100406@alu.cdut.edu.cn (X.C.); shixinyi@stu.cdut.edu.cn (X.S.); fengziwei@stu.cdut.edu.cn (Z.F.); mengyichen@stu.cdut.edu.cn (Y.M.)
2    Sichuan Institute of Geological Survey, Chengdu 610081, China; ypldxm@163.com
*    Correspondence: pengjunsheng@cdut.edu.cn (J.P.); chenwende@cdut.edu.cn (W.C.)

**Abstract:** Spatial interpolation is a crucial aspect of soil toxic element pollution research, serving as a vital foundation for pollution assessment, treatment, and sustainability efforts. The selection and adjustment of interpolation methods directly influences the accuracy of spatial distribution maps and data results, thereby indirectly impacting related research. This paper conducts a comparative study of different interpolation methods and analyses the sources of soil toxic elements in the study area of Cangxi County, aiming to provide a scientific foundation for future soil management, remediation, and enhanced local sustainability. The spatial correlation of As, Cd, Hg, Mn, Pb, and Mo in 228 surface soil samples in the study area of Cangxi County is analyzed. The interpolation results, spatial distribution of OK (ordinary Kriging), IDW (inverse distance weighting), RBF (radial basis function) and the changes of pollution area after interpolation are compared. The smoothing effect is assessed based on the comparison results, interpolation accuracy, and impact on pollution assessment of OK, IDW, and RBF. The interpolation method most suitable for each metal in the study area is selected. It can be concluded that the optimal interpolation method for As, Hg, and Mn is IDW; for Cd and Mo, it is RBF; and for Pb, it is OK. After the correlation analysis of toxic elements in the soil of the study area, the PMF (positive matrix factorization) model and hotspot analysis is applied to analyzing the source of toxic elements. The analysis indicates that the predominant sources of pollution are anthropogenic, categorized into industrial activities (30.8%), atmospheric deposition caused by coal combustion and traffic exhaust (21.5%) and agricultural activities (19.5%). Natural sources, such as soil parent material, contribute to 28.2% of the pollution on average.

**Keywords:** spatial interpolation; OK (ordinary kriging); IDW (inverse distance weighting); RBF (radial basis function); interpolation accuracy; smoothing effect; pollution source analysis; PMF model

## 1. Introduction

Soil plays a critical role in sustainability, impacting agricultural, economic, and human health aspects. Moreover, soil pollution by toxic elements is a significant environmental concern worldwide [1], garnering considerable attention [2]. The latest soil pollution survey in China revealed that 16.1% of samples surpassed the environmental quality standards set by the Ministry of Environmental Protection, with toxic elements constituting 82.4% of the pollutants [3]. Pollution from toxic elements differs from organic pollution; it is non-biodegradable [4], and can readily accumulate in the bodies of humans and animals through the food chain, posing health risks [5,6]. Soil pollution by toxic elements negatively affects local sustainability, contaminating farm crops, undermining the agricultural economy, and increasing health risks for local residents.

Spatial interpolation, a process that estimates the value of a certain attribute at a given position based on values sampled at adjacent points, plays a crucial role in researching soil toxic element pollution and in environmental management and conservation [7]. Obtaining a spatial distribution map with high accuracy of soil toxic elements by spatial interpolation is essential for early warning of soil pollution and further enhancing local sustainability [8]. Common spatial interpolation methods in geostatistics, such as Kriging [9], IDW (inverse distance weight) [10], RBF (radial basis function) [11], and LPI (local polynomial) [12], are widely used in soil pollution research and soil pollution spatial distribution mapping [13–15]. At the same time, related studies have been conducted to compare the outcomes of various spatial interpolation methods, aiming to identify the most accurate and suitable method for specific study areas. Fu et al. (2014) applied four spatial interpolation methods to interpolate soil toxic elements in Lishui District, Nanjing City, China. The smoothing effect was evaluated by comparing the interpolation results, and then the optimal interpolation parameters and methods were selected [16]. Sheng et al. (2020) employed IDW and RBF to interpolate the Cd elements sampled at different soil depths in the lower reaches of the Fujiang River, China, and determined the optimal interpolation method suitable for soil samples at different depths by assessing interpolation accuracy and spatial distribution maps [17]. Chen et al. (2022) used OK, IDW, RBF, and LPI to interpolate As, Cu, and Mn in the soil of Chongqing, China with the optimal parameters, and assessed the interpolation results for spatial heterogeneity analysis and source analysis of soil toxic elements in Chongqing, which made great contributions to local soil sustainability [18]. In summary, selecting appropriate interpolation methods and optimizing parameters are crucial for achieving high-accuracy interpolations in related research, thereby fostering sustainable development.

Meanwhile, analyzing the sources of toxic element pollution is crucial for soil protection and remediation [19]. This analysis serves as a vital reminder to control soil pollution at its source, significantly contributing to enhancing local soil sustainability. It is generally believed that soil toxic elements originate from two main sources: natural and anthropogenic. Natural sources, often minor and primarily resulting from the weathering of parent rocks, contrast with the larger and more complex anthropogenic sources. These man-made sources, fueled by various human activities, significantly contribute to increased pollution levels [20]. The sources of soil toxic elements are primarily analyzed using receptor models, including the following: PMF (positive matrix factorization method), Unmix model, APCS-MLR (absolute factor score-multiple linear regression method), and so on [21–23]. Among these models, the PMF (positive matrix factorization) has been highlighted for its convenience and efficiency. Additionally, its analysis results are noted for their accuracy [22,24–26].

The study area is the key planting area of red kiwifruit and walnut planned by the Cangxi government. In the past, the researches on soil toxic element pollution in Cangxi County were limited to pollution evaluation [27], and there was no discussion on the selection of spatial interpolation methods and source analysis. Therefore, this study employs OK, IDW, and RBF methods to interpolate the presence of As, Cd, Hg, Mn, Pb, and Mo in soil of the study area, with optimal parameters. The results of this interpolation were evaluated based on their accuracy, result statistics, spatial distribution, and impact on pollution assessment, leading to the selection of the best interpolation method for each soil toxic element included. Concurrently, the PMF model was utilized to analyze the source of the toxic element pollution, providing a scientific foundation for future pollution control and soil remediation efforts. These findings will enhance local agricultural, economic, and human sustainability in the study area.

## 2. Materials and Methods

### 2.1. Overview of the Study Area

The study area (31°51′ N~32°55′ N, 105°57′ E~106°02′ E) is at the junction of Sanchuan Town, Baihe Town, and Shimen Town in Cangxi County, Guangyuan City, and Sichuan Province, with an acreage of 14.88 km² (Figure 1). The study area is a purple hilly area, the terrain is tilted from northwest to southeast, and the slope is gentle. The study area soil is dominated by purple soil, which accounts for 70% of the total area, while paddy soil accounts for 30%. The main crop is kiwifruit. Cangxi County belongs to the subtropical humid monsoon climate zone. The average annual temperature is 16.9 °C, the average temperature in January is 6 °C, and in July it is 27 °C. The extreme minimum temperature is −4.6 °C, while the maximum temperature is 39.3 °C. The diurnal amplitude is 3~7 °C. The annual frost-free period lasts for 288 days, and the average annual rainfall is about 1100 mm.

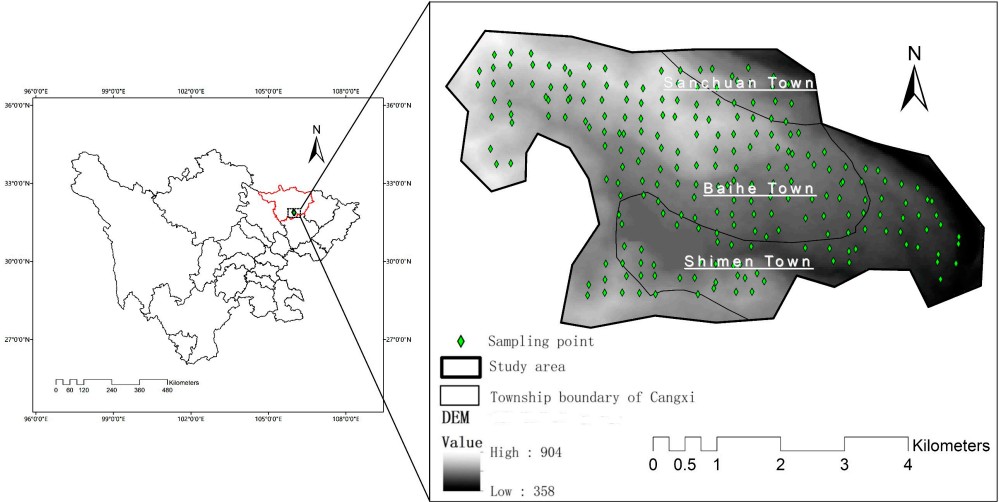

**Figure 1.** Study area location and surface soil sampling point diagram.

### 2.2. Sample Collection and Analysis

A total of 228 topsoil (cultivated layer) samples at an average density of 16 pieces/km² were collected. GPS was used to navigate to the pre-set sampling point and measure its geographical coordinates. Each sampling point was marked, numbered, and documented with corresponding landscape photographs. The sampling tool used was a stainless-steel dredge shovel. The sampling medium was the soil column of 0~20 cm on the surface, and the soil column combination of 3~5 points (radial) was collected within the radius of 50 m around the sampling point as a sample. Efforts were made to minimize soil disruption to ensure that the samples were not secondary polluted during collection. The samples were stored in designated sample bags and isolated using a polyester bag.

In this study, the soil sample analysis was undertaken by the Chengdu Mineral Resources Supervision and Inspection Center of the Ministry of Land and Resources. The analysis unit strictly adheres to the "Technical standard of geological survey of China Geological Survey (DD2005-01~DD2005-03)" published by China Geological Survey in October 2005 and rigorously monitors the analysis quality of various samples through the use of standard samples, internal laboratory inspections, cryptographic inspections, and external sampling inspections. The detection center selected the full amount of soil sample elements. After being received by the detection center, air-dried, and gravel and plant debris were removed, the sample was divided into two parts. One part was ground with agate mortar, passed through a soil sieve with a diameter of 2 mm, and then ground all through a 100-mesh sieve. Different spectroscopic analysis methods were used to determine the toxic element concentration of samples. The spectroscopic analysis methods, analytical equipment and method detection limits (MDL) of each toxic element were measured are

shown in Table 1. The X series II American thermoelectric plasma mass spectrometer and Perform' X4200 X-ray fluorescence spectrometer is manufactured by Thermo Feld in the US. The AFS-2998 atomic fluorescence photometer and AFS-2999 atomic fluorescence photometer are produced by Beijing Beifang Rayleigh Analytical Instruments (Group) Co., LTD in China (Beijing, China). The standards for the selection of determination methods are based on the following: "Soil and sediment-Determination of mercury, arsenic, selenium, bismuth, antimony-Microwave dissolution/Atomic Fluorescence Spectrometry (HJ-680 2013)" published by Ministry of Environmental Protection (China) on 21 November 2013, for As and Hg determination using AFS; "Soil and sediment-Determination of inorganic element-Wavelength dispersive X-ray fluorescence spectrometry (HJ-780 2013)" published by Ministry of Environmental Protection (China) on 21 November 2013, for Mn and Pb determination using XRF; and "Soil and sediment-Determination of aqua regia extracts of 12 metal elements-Inductively coupled plasma mass spectrometry (HJ-803 2016)" published by Ministry of Environmental Protection (China) on 24 June 2016, for Cd and Mo determination using ICP-MS. Another part of the sample was for pH determination and direct sunlight, acid, alkali, other gases, and dust pollution were avoided. At the same time, another spare part of the sample was extracted by quartering method, the debris other than soil was removed, and the test screen with 0.84 mm aperture was used to filter. Potentiometry was used to determine pH value.

**Table 1.** Spectroscopic analysis used to determine the total content of each toxic element in soil and their respective method detection limits (MDL).

| Element | Spectroscopic Analysis | Analytical Equipment | Method Detection Limits (MDL, mg·kg$^{-1}$) |
|---|---|---|---|
| As | AFS | AFS-2999 atomic fluorescence photometer | 0.5 |
| Cd | ICP-MS | X series II American thermoelectric plasma mass spectrometer | 0.05 |
| Hg | AFS | AFS-2998 atomic fluorescence photometer | 0.003 |
| Mn | XRF | Perform' X4200 X-ray fluorescence spectrometer | 10 |
| Mo | ICP-MS | X series II American thermoelectric plasma mass spectrometer | 0.25 |
| Pb | XRF | Perform' X4200 X-ray fluorescence spectrometer | 2 |

### 2.3. Analysis Methods

Microsoft Excel 2013 and SPSS (Statistical Package for the Social Sciences) 27.0 was used to analyze the data. ArcGIS 10.7 was used to proceed geological statistical analysis. OK, IDW, and RBF interpolation were also conducted with ArcGIS. PMF was analyzed using EPA PMF 5.0. The methods and formulas employed in the data analysis are as follows:

The coefficient of variation (CV) is an index that assesses the dispersion degree of a single factor in spatial distribution [28,29]. The higher the CV, the stronger the interference from human activities or the greater the degree of pollution [30–33]. The coefficient of variation can be categorized into weak variation (CV < 15%), moderate variation (15% ≤ CV ≤ 36%) and strong variation (CV > 36%) [34].

Kriging, widely used in geostatistics, is the most usual interpolation method [35]. It is an unbiased, optimal estimation method for regionalized variables in a confined area, based on variogram theory and structural analysis [36,37]. Its main advantages include not only predictive results but also error estimation and the spatial autocorrelation of soil characteristics, interpolated from known to estimated points. It exhibits excellent intrinsic correlation attributes and accuracy, beneficial for evaluating the uncertainty of prediction results [35,37]. Ordinary Kriging (OK) was used in this study.

Inverse distance weighting (IDW) is an interpolation method related to spatial distance [22]. It operates on the principle of similarity that the closer two objects are, the more similar they are likely to be; conversely, the greater the distance between them, the less similar they are. IDW uses the distance between the interpolation point and the sample point as a weight, assigning greater weight to sample points that are closer to the interpolation point [16].

The radial basis function method (RBF) is a variation function model. It calculates a set of weight coefficients of the nodes to be estimated by the base function and adjusts the smoothing factor in the base function to control the smoothness of the interpolation surface and the estimation accuracy, so as to achieve smooth interpolation. This method is suitable for interpolation of surfaces with gentle surface changes, which is susceptible to the influence of maximum and minimum values, and can predict the values of unknown points higher or lower than the sample points [35].

To compare interpolation methods, it is essential to minimize the original data's error, and an effective strategy for error reduction is to optimize parameters during the interpolation process for optimal results [38]. Analyzing the interpolation outcomes for various toxic elements with different parameters in ArcGIS reveals that using a combination of parameters which yields the least inaccuracy is beneficial. By adjusting the range of each toxic element in the interpolation process to ensure it is sufficiently large, and by keeping the nugget coefficient ($C_o/(C_o + C)$) small, interpolation can achieve a broader spatial autocorrelation range and stronger spatial autocorrelation, thereby enhancing accuracy [39].

In cross validation, performing linear regression analysis helps achieve the algorithm's goal of unbiased mean value prediction. The measured values of sampling points were used as independent variables, and the predicted values of cross-validation were used as dependent variables. Before the intersection of the linear model and the 1:1 straight line, the predicted value is greater than the measured value, while the result is opposite after the intersection [40,41].

The commonly used evaluation indexes for cross-validation include *RMSE* (Root Mean Square Error) and *ME* (Mean Error), among others. *RMSE* serves as a metric for assessing the accuracy of predictions. A smaller *RMSE* value indicates a higher level of precision in the interpolation. ME is the measure of prediction accuracy. *ME* serves as a metric for assessing the impartiality of predictions, with values closer to 0 indicating a higher degree of unbiasedness in the interpolation [39]. Their expressions are as follows:

$$RMSE = \sqrt{\frac{1}{n} \sum_{i=1}^{n} V_i^2} \tag{1}$$

$$ME = \frac{1}{n} \sum_{i=1}^{n} V_i^2 \tag{2}$$

where $V_i$ is the error between the measured value and the predicted value.

*IP* (Imprecision) is the variation of prediction error. The smaller the *IP* is, the more accurate the interpolation result is [30]. Its expression is as follows:

$$IP^2 = RMSE^2 - ME^2 \tag{3}$$

*RI* (the potential ecological risk index) is widely used to assess the degree of soil toxic element pollution. It introduces the toxicity coefficient of toxic elements, links the environmental ecology with the toxicological effects of toxic elements, and considers various factors for analysis [31]. Its expression is as follows:

$$C_f^i = C^i / C_n^i \tag{4}$$

$$E_r^i = T_r^i \times C_f^i \tag{5}$$

$$RI = \sum E_r^i \ \sum T_r^i \times C_f^i \tag{6}$$

where $C_f^i$ is the $i$-th toxic element's enrichment coefficient; $C^i$ is the $i$-th toxic element's measured concentration; and $C_n^i$ is the $i$-th toxic element's assessment standard, which is the $i$-th toxic element's soil background value of Chengdu. $T_r^i$ is the $i$-th toxic element's toxicity coefficient. Toxicity coefficients (Tr) of the examined elements are shown in Table 2.

RI is the potential ecological risk index of multi-element environment in soil; $E_r^i$ is the $i$-th toxic element's potential ecological risk index [42,43].

**Table 2.** Toxicity coefficients (Tr) of the examined elements [42].

| Element | As | Cd | Hg | Mn | Pb | Mo |
|---|---|---|---|---|---|---|
| Toxicity coefficient | 10 | 30 | 40 | 1 | 5 | 15 |

The potential ecological risk index $E_r^i$, RI classification and its corresponding pollution degree are shown in Table 3.

**Table 3.** Potential ecological risk classification standard [44].

| $E_r^i$ | Single Factor Ecological Risk Pollution Degree | RI | Total Potential Ecological Risk Pollution Degree |
|---|---|---|---|
| $E_r^i < 40$ | Light | $RI < 150$ | Light |
| $40 \leq E_r^i < 80$ | Moderate | $150 \leq RI < 300$ | Moderate |
| $80 \leq E_r^i < 160$ | Strong | $300 \leq RI < 600$ | Strong |
| $160 \leq E_r^i < 320$ | Very strong | $RI \geq 600$ | Extremely strong |
| $E_r^i \geq 320$ | Extremely strong | | |

The positive matrix factor analysis (PMF) is a method for quantitative analysis of multivariate factors of pollution sources by using sample composition or fingerprints through the mathematical method of receptor model [45]. The operation principle of PMF model is as follows:

$$X_{ij} = \sum_{k=1}^{P} g_{ik} f_{ki} + e_{ij} \tag{7}$$

where $i$ is the number of samples; $j$ is the number of elements; $X_{ij}$ is the $j$-th toxic element concentration in the $i$-th sample; $g_{ik}$ is the relative contribution of source $k$ to sample $i$; $f_{ki}$ is the content of element j on the source; and $e_{ij}$ is the residual matrix.

The optimal matrices G and F are obtained by decomposing the original matrix X of the PMF model, so that the objective function $Q$ is minimized. The calculation of the objective function $Q$ is as follows:

$$Q = \sum_{i}^{n} \sum_{j=1}^{m} \left( E_{ij} / U_{ij} \right)^2 \tag{8}$$

where $U_{ij}$ is the uncertainty of the content of the $j$-th element in the $i$-th sample. The model can weight each individual data point and give each data point the appropriate uncertainty. When the concentration of the element is lower than or equal to the corresponding method detection limit (MDL), the uncertainty is calculated as Formula (9), otherwise the uncertainty is calculated as Formula (10).

$$U_{ij} = \partial/10 + MDL/3 \tag{9}$$

$$U_{ij} = \sqrt{(\delta \times \partial)^2 + MDL^2} \tag{10}$$

where $\delta$ is the relative standard deviation; and $\partial$ is the concentration of chemical elements [46]. The method detection limit (MDL) of toxic element elements involved in this study is shown in Table 1.

## 3. Results

### 3.1. Descriptive Statistics of Soil Toxic Element Content

The experimental results of this study are presented in Table 4. The soil pH value was between 5.26 and 8.56. The CV of Hg in the study area is 46.545%, suggesting a strong level of variation, which implies that Hg pollution is more likely related to human activities. The

CV of Pb is 12.523%, indicating a weak variation and suggesting that Pb pollution is less likely to be related to human activities. Other toxic elements are moderately variable and related to human activities.

**Table 4.** Statistical results of soil toxic elements contents and reference soil background values for China and Chengdu area.

| Element | | As | Cd | Hg | Mn | Mo | Pb | pH |
|---|---|---|---|---|---|---|---|---|
| Max (mg·kg$^{-1}$) | | 20.200 | 0.589 | 0.142 | 861.000 | 1.240 | 33.300 | 8.56 |
| Min (mg·kg$^{-1}$) | | 3.100 | 0.058 | 0.012 | 211.000 | 0.272 | 11.500 | 5.26 |
| Median (mg·kg$^{-1}$) | | 9.305 | 0.228 | 0.041 | 486.500 | 0.508 | 24.900 | 7.33 |
| Mean (mg·kg$^{-1}$) | | 9.735 | 0.223 | 0.043 | 494.798 | 0.554 | 24.756 | 7.26 |
| SD | | 2.330 | 0.066 | 0.020 | 121.866 | 0.176 | 3.100 | 0.92 |
| CV (%) | | 23.931 | 29.485 | 46.545 | 24.629 | 31.783 | 12.523 | 12.76 |
| Chinese soil background value [47] | | 11.200 | 0.097 | 0.065 | 583.000 | 2.000 | 26.000 | — |
| Chengdu soil background value [48] | | 13.000 | 0.130 | 0.047 | 852.000 | 0.600 | 23.000 | — |
| Soil Environmental Quality Standard [1] | pH ≤ 5.5 | 40 | 0.3 | 1.3 | — | — | 70 | — |
| | 5.5 < pH ≤ 6.5 | 40 | 0.3 | 1.8 | — | — | 90 | — |
| | 6.5 < pH ≤ 7.5 | 30 | 0.3 | 2.4 | — | — | 120 | — |
| | pH > 7.5 | 25 | 0.6 | 3.4 | — | — | 170 | — |

[1] The data comes from "Soil environmental quality Risk control standard for soil contamination of agricultural land (GB 15618-2018) [49]" published by the Ministry of Ecology and Environment (China) on 22 June 2018.

*3.2. Assessment of Toxic Element Pollution in Soil*

The results of the potential ecological risk index method are shown in Table 5. From the average value of $E_r^i$, the potential ecological risk of the surface layer in the study area, from light to moderate, is as follows: Mn (0.581) < Pb (5.382) < As (7.489) < Mo (13.848) < Hg (36.827) < Cd (51.454). According to the classification criteria in Table 3, Cd in the surface soil reaches moderate pollution, and As, Hg, Mn, Pb, and Mo are all lightly polluted in the study area. From the RI value, the soil of the study area is generally at a light pollution level.

**Table 5.** Statistical analysis of soil toxic element content.

| $E_r^i$ | As | Cd | Hg | Mn | Pb | Mo | *RI* |
|---|---|---|---|---|---|---|---|
| Max | 15.538 | 135.923 | 120.851 | 1.011 | 7.239 | 31.000 | 222.650 |
| Min | 2.385 | 13.385 | 10.213 | 0.248 | 2.500 | 6.800 | 57.087 |
| Median | 7.158 | 52.500 | 34.468 | 0.571 | 5.413 | 12.700 | 114.751 |
| Mean | 7.489 | 51.454 | 36.827 | 0.581 | 5.382 | 13.848 | 115.581 |

To further analyze the pollution status of each toxic element, the frequency distribution of $E_r^i$ and *RI* was calculated and counted. The results are shown in Table 6. For all soil samples, As, Mn, Pb, and Mo are entirely at a light potential ecological risk level; For Cd, 21.93% of the sampling points are at light pollution level, 75.44% of the sampling points are at moderate pollution level, and 2.63% of the sampling points are at strong pollution level. For Hg, 58.77% of the sampling points are at light pollution level, 39.04% of the sampling sites are at moderate pollution level, and 2.19% of the sampling sites are at strong pollution level. From RI, 92.54% of the sampling points are at light pollution level, and 7.46% of the sampling points are at moderate pollution level.

**Table 6.** Frequency distribution of $E_r^i$ and *RI* in soil.

| Ecological Risk Pollution Degree | As | Cd | Hg | Mn | Pb | Mo | RI |
|---|---|---|---|---|---|---|---|
| light | 100.00 | 21.93 | 58.77 | 100.00 | 100.00 | 100.00 | 92.54 |
| moderate | 0.00 | 75.44 | 39.04 | 0.00 | 0.00 | 0.00 | 7.46 |
| Strong | 0.00 | 2.63 | 2.19 | 0.00 | 0.00 | 0.00 | 0.00 |
| Very strong | 0.00 | 0.00 | 0.00 | 0.00 | 0.00 | 0.00 | |
| Extremely strong | 0.00 | 0.00 | 0.00 | 0.00 | 0.00 | 0.00 | 0.00 |

### 3.3. Parameter Optimization of Different Interpolation Methods for Soil Toxic Elements

Through experiments, the optimal parameters of the variogram theoretical model of OK method are shown in Table 7. The optimal theoretical model for As, Mn, Pb, and Mo is exponential. Spherical is the optimal theoretical model for Cd. Gaussian is optimal for Hg.

**Table 7.** The theoretical model of variation function of soil toxic element elements and its related parameters.

| Element | Theoretical Model | Nugget ($C_o$) | Sill ($C_o + C$) | Range (m) | $C_o/(C_o + C)$ | $R^2$ | RSS |
|---|---|---|---|---|---|---|---|
| As | Exponential | 4.390 | 466.279 | 9.300 | 0.009 | 0.385 | $-4.70 \times 10^{-15}$ |
| Cd | Spherical | 0.003 | 0.198 | 0.940 | 0.016 | 0.554 | $1.05 \times 10^{-16}$ |
| Hg | Gaussian | 0.000 | 0.000 | 0.002 | 2.580 | 0.648 | $-2.16 \times 10^{-17}$ |
| Mn | Exponential | 13,981.034 | 409,876.614 | 18.000 | 0.034 | 0.166 | $5.81 \times 10^{-14}$ |
| Pb | Exponential | 8.283 | 671.014 | 16.330 | 0.012 | 0.317 | $-6.79 \times 10^{-15}$ |
| Mo | Exponential | 0.028 | 1.520 | 38.290 | 0.019 | 0.265 | $-1.79 \times 10^{-16}$ |

The optimal parameters of IDW and RBF are presented in Table 8. Through experiments, it is found that when the power value in the interpolation process of each element is 1, the inaccuracy is relatively minimal. In the RBF experiment, As, Hg, Mn, and Pb were interpolated using the inverse multiquadric function, while the Cd and Mo elements were interpolated using the spline with tension function.

**Table 8.** The optimal parameters of IDW and RBF for soil toxic element elements.

| Interpolation Method | Parameter | As | Cd | Hg | Mn | Pb | Mo |
|---|---|---|---|---|---|---|---|
| IDW | Power | 1 | 1 | 1 | 1 | 1 | 1 |
| RBF | Kernel function | Inverse Multiquadric | Spline with Tension | Inverse Multiquadric | Inverse Multiquadric | Inverse Multiquadric | Spline with Tension |
| | Parameter | $1.36 \times 10^{-5}$ | 48,660.323 | $2.42 \times 10^{-5}$ | $1.18 \times 10^{-38}$ | $6.40 \times 10^{-5}$ | 48,660.323 |

### 3.4. Accuracy Comparison of Different Interpolation Methods for Soil Toxic Elements

The cross-validation method was used to assess the accuracy of the interpolation method. The regression line intersected with the 1:1 straight line, as shown in Figure 2. The correlation coefficient between the predicted and the measured values of As and Hg by IDW is the largest, followed by RBF and OK. The correlation coefficient of Cd, Pb, and Mo by OK is the smallest, significantly lower than the similar coefficients with IDW and RBF. The correlation coefficient of Mn is highest using IDW, and with RBF it is slightly lower than with OK. The correlation coefficient using OK ranges from 0.1170 to 0.3997. The correlation coefficient using IDW ranges from 0.8030 to 0.8388. The correlation coefficient by RBF ranges from 0.0992 to 0.8388. In general, the fitting degree of OK is significantly lower than that of IDW and RBF, and its correlation coefficient is also generally smaller than that of the other two methods.

The *RMSE*, *ME*, and *IP* values were computed to assess the accuracy of the three interpolation methods. The results are presented in Table 9. For As interpolation, OK yields an *ME*

closest to 0 and has the smallest *IP*, whereas RBF achieves the smallest *RMSE*. For Cd interpolation, RBF's *ME* is closest to 0, with OK and IDW showing similar results, and relatively low *RMSE* and *IP* values. When OK is used to interpolate Hg, it results in an *ME* closest to 0, with its *RMSE* and *IP* values being slightly smaller than those achieved by RBF and IDW. For Mn interpolation, IDW's *ME* is closest to 0, with the smallest *RMSE* and *IP* observed when using OK method. Pb interpolation conducted by OK is remarkably accurate, with *ME* near 0 and the lowest *RMSE* and *IP* values. OK's *ME* for Mo interpolation approaches 0, while RBF and IDW present similar and relatively low *RMSE* and *IP*. On average, the *ME* generated by IDW is relatively close to 0, and the *RMSE* and *IP* produced by OK are comparatively small.

**Table 9.** *RMSE*, *ME*, and *IP* of three interpolation methods for soil toxic elements.

|  | OK | | | IDW | | | RBF | | |
|---|---|---|---|---|---|---|---|---|---|
|  | ME | RMSE | IP | ME | RMSE | IP | ME | RMSE | IP |
| As | $-0.012$ | 2.208 | 2.208 | 0.013 | 2.259 | 2.259 | 0.027 | 2.188 | 2.188 |
| Cd | $4.11 \times 10^{-4}$ | 0.067 | 0.067 | $2.50 \times 10^{-4}$ | 0.067 | 0.067 | $3.10 \times 10^{-6}$ | 0.068 | 0.068 |
| Hg | $-5.16 \times 10^{-5}$ | 0.021 | 0.021 | $6.90 \times 10^{-4}$ | 0.020 | 0.020 | $9.90 \times 10^{-4}$ | 0.020 | 0.020 |
| Mn | $-4.125$ | 120.945 | 120.875 | $-2.865$ | 125.237 | 125.204 | $-4.025$ | 122.966 | 122.900 |
| Pb | 0.038 | 2.891 | 2.891 | 0.071 | 2.956 | 2.955 | 0.082 | 2.933 | 2.932 |
| Mo | $3.65 \times 10^{-4}$ | 0.163 | 0.163 | $2.85 \times 10^{-3}$ | 0.162 | 0.162 | $1.52 \times 10^{-3}$ | 0.162 | 0.162 |
| Mean | $-0.683$ | 21.049 | 21.037 | $-0.463$ | 21.784 | 21.778 | $-0.652$ | 21.390 | 21.378 |

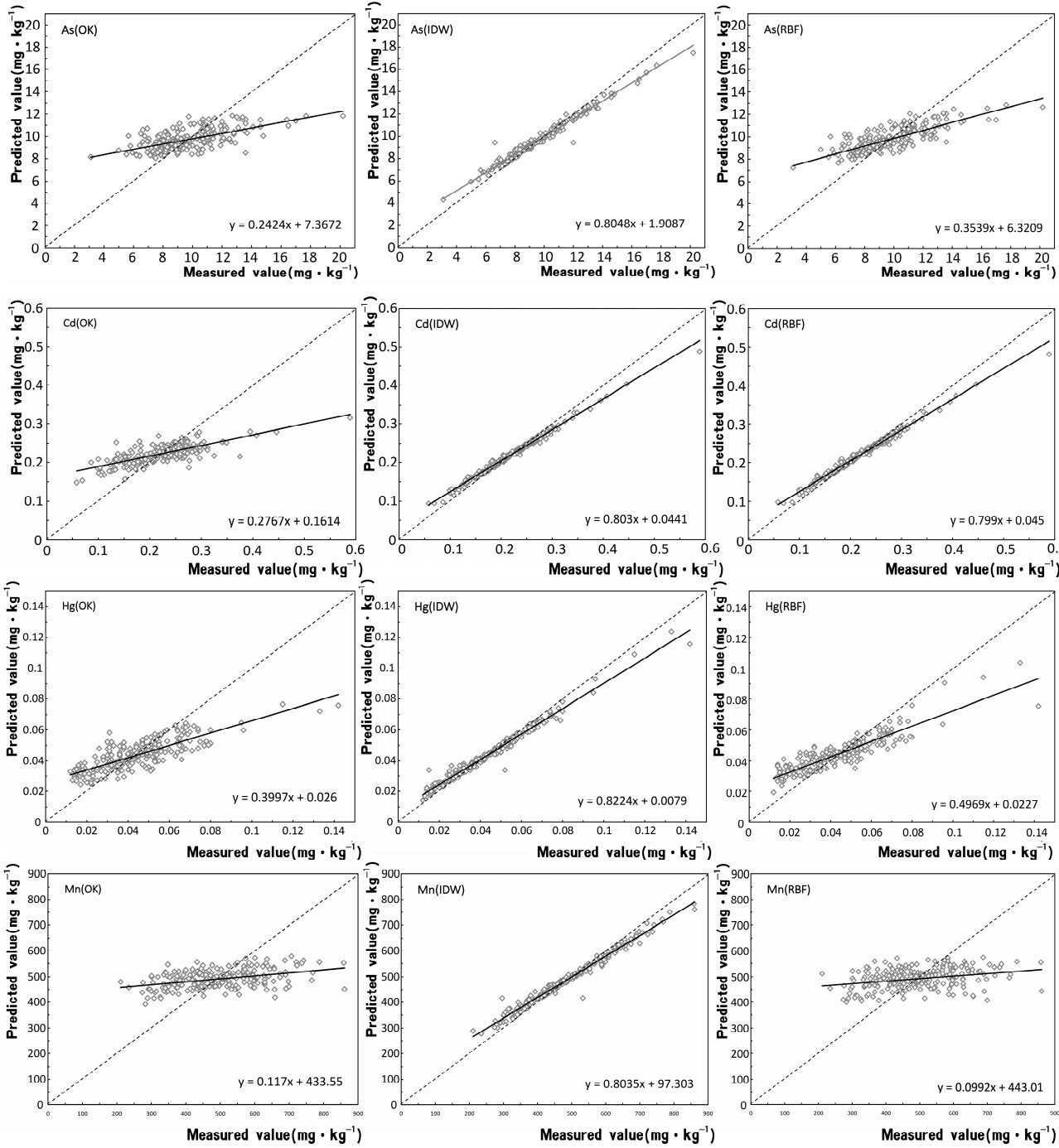

**Figure 2.** *Cont.*

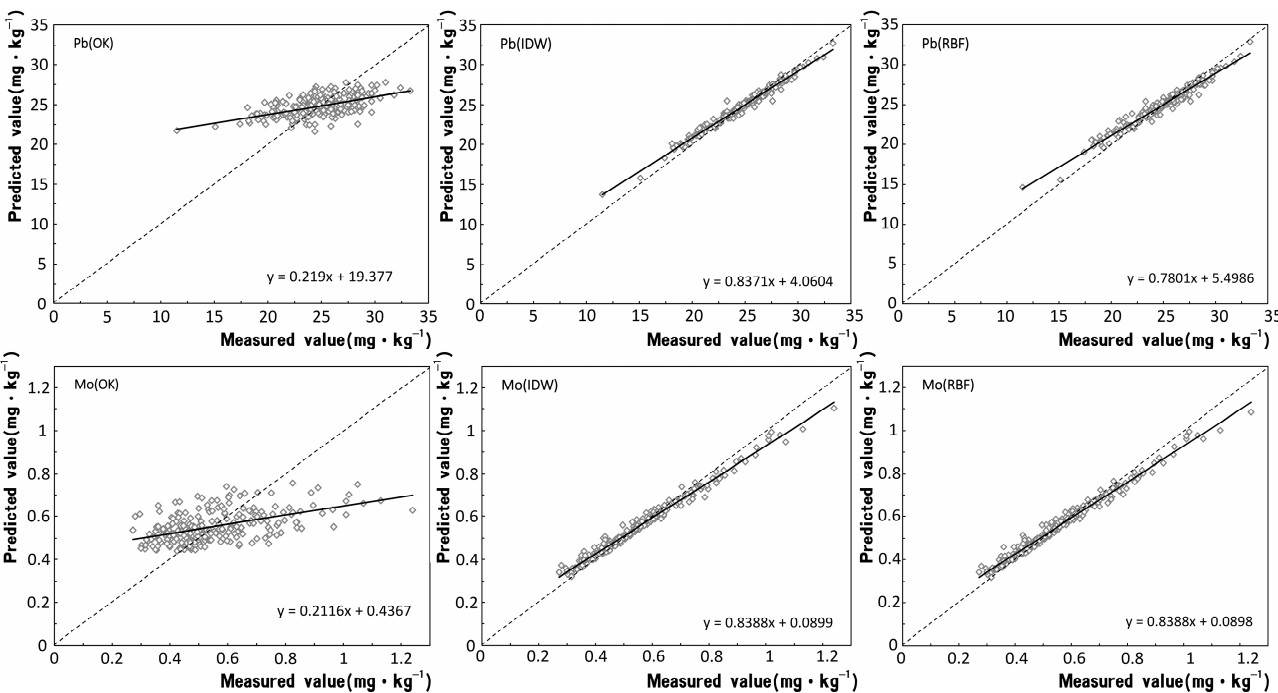

**Figure 2.** Cross-validation of three interpolation methods for soil toxic elements.

*3.5. Comparison of Interpolation Results' Statistics and Spatial Distribution of Different Interpolation Methods for Soil Toxic Elements*

The maximum, minimum, mean value, SD (standard deviation) and CV (coefficient of variation) of the results by the three interpolation methods were statistically analyzed and compared with the relevant data of the sampling points. The results are presented in Table 10. The smoothing effect of the three interpolation methods impacts the measured concentration range of toxic elements and the intensity of the smoothing effect can be explained by the impact amplitude. In general, the smoothing effects of IDW and RBF are significantly weaker than that of OK. The concentration range of toxic elements showed different degrees of reduction after interpolation, OK had the largest reduction, followed by RBF, while IDW had the smallest reduction. The mean values of the six toxic element elements after different interpolation methods did not change significantly. The average value increased for all elements except Cd after IDW interpolation, while the average values of the other elements decreased by varying degrees after different interpolation methods were applied. Except for Cd and Mo interpolated by IDW, which are closest to the measured values, the mean values of other elements obtained by RBF are the closest to the sampling points, followed by IDW, with OK showing the largest discrepancy. After interpolation, both SD and CV significantly reduced, a result of the smoothing effect. The SD and CV of Cd and Hg are the closest to the sampling point after OK interpolation. For As, Mn, and Pb, the smallest SD and CV values occur after IDW interpolation, while those for Mo are closest to the sampling point after RBF interpolation.

The content distribution maps of different toxic element elements were generated using various spatial interpolation methods in ArcGIS, and the results are presented in Figure 3. The soil content of As is higher in the northern part of the study area and at the junction of Baihe Township and Shimen Township in the central part. When comparing the three methods for the prediction of low concentration (3.100~8.762 mg·kg$^{-1}$) of the As element, it is found that OK is rougher, while IDW provides more detail for a higher concentration range (9.657~20.200 mg·kg$^{-1}$). The Cd soil content in the study area is higher in the northern Sanchuan Town and the central and southern Baihe Township. All three interpolation methods have a "bull's eye" phenomenon near the sampling points with higher content, potentially indicating point source pollution and a small contaminated

area due to concentrations near the relevant sampling points exceeding standards. The prediction of OK for higher concentrations (0.235~0.589 mg·kg$^{-1}$) is significantly rougher, whereas the prediction of RBF for both higher and lower concentrations is relatively more accurate. The Hg content is higher at the junction of Baihe Township and Shimen Township in the central part of the study area. The results of RBF interpolation are significantly rougher than the other two methods, while OK provides better predictions than IDW. The Mn content is mainly concentrated in the soil at the junction of Baihe Township and Shimen Township in the south of the study area, in Baihe Township to the east, and in Sanchuan Township to the north. For predicting Mn, IDW performs significantly better than the other two methods for both higher and lower concentration ranges. The soil Pb content at the junction of Baihe Township and Shimen Township in the middle of the study area and Baihe Township in the north is relatively high. For the prediction of Pb, the three methods yield similar results, with the OK method slightly outperforming the other two from a detailed perspective. The soil content of Mo is mainly concentrated along the border between Baihe Township and Shimen Township in the middle of the study area and in Baihe Township to the north. Although the prediction of OK for lower concentration (0.272~0.467 mg·kg$^{-1}$) is more detailed, compared to the other two methods, its prediction for medium and high concentration (0.534~1.240 mg·kg$^{-1}$) is less accurate, with RBF being the best for predicting medium and high concentrations.

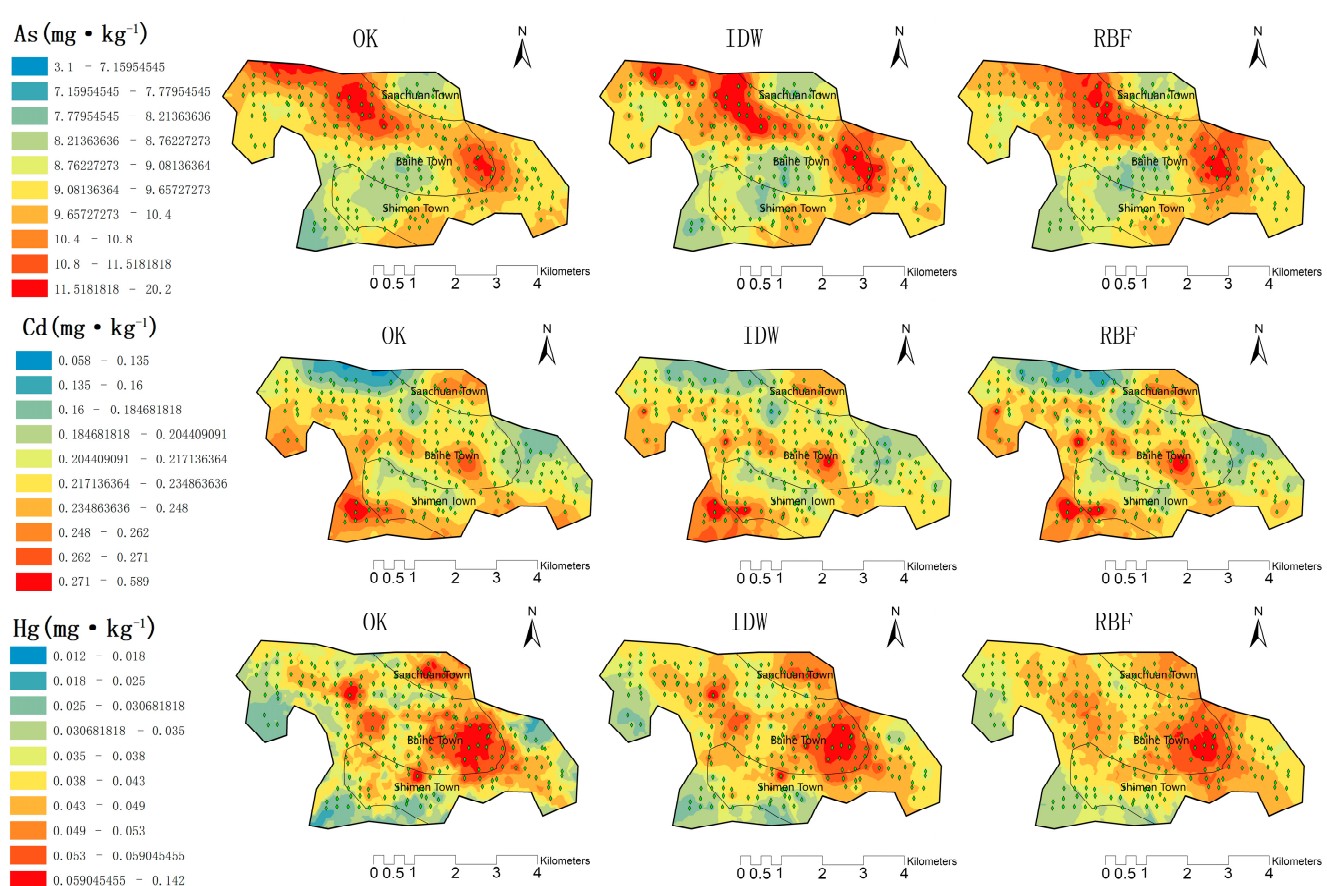

**Figure 3.** *Cont.*

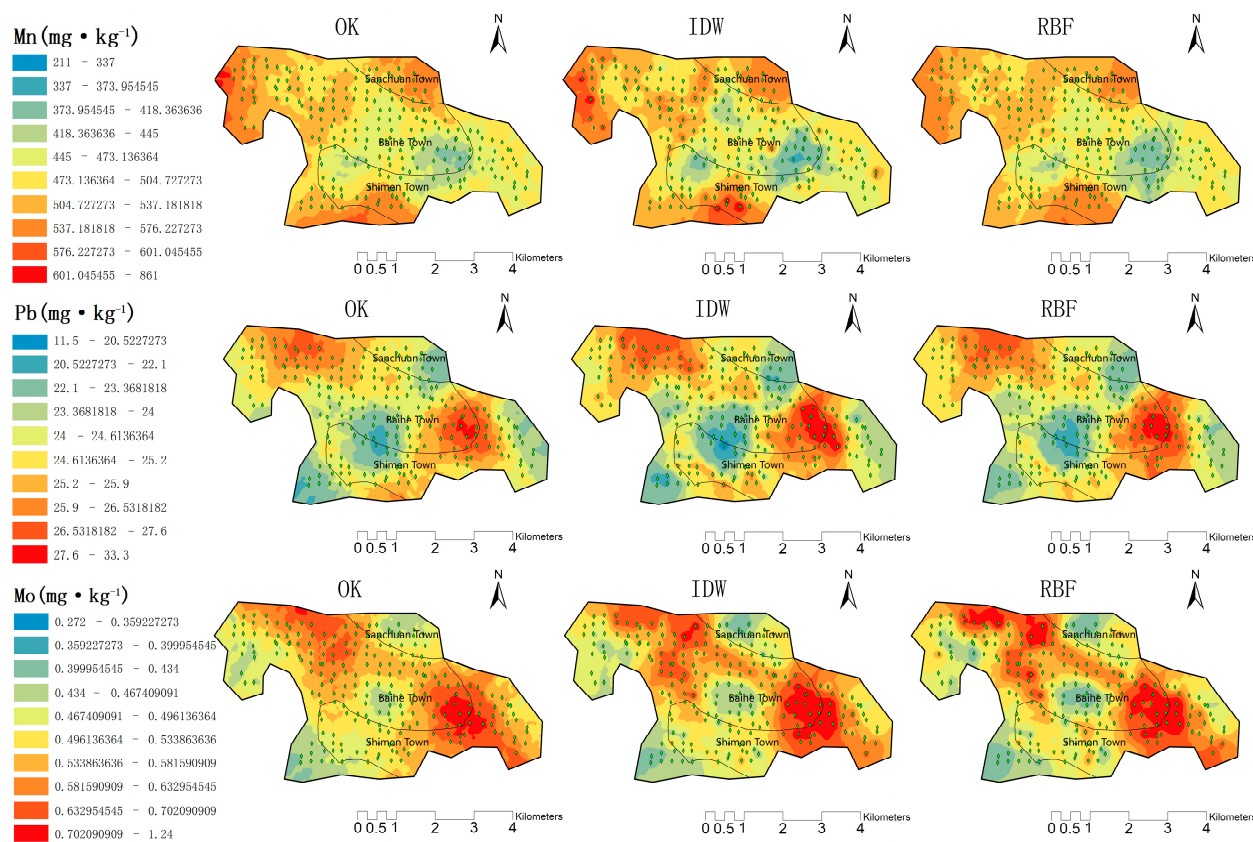

**Figure 3.** Comparison of spatial distribution of soil toxic elements after interpolation by three interpolation methods.

**Table 10.** Statistics of interpolation results of different interpolation methods for toxic elements in soil.

| Element | Method | Min (mg·kg$^{-1}$) | Max (mg·kg$^{-1}$) | Mean (mg·kg$^{-1}$) | SD | CV (%) |
|---|---|---|---|---|---|---|
| As | Sampling point | 3.100 | 20.200 | 9.735 | 2.330 | 23.931 |
| | OK | 7.505 | 12.721 | 9.476 | 0.851 | 8.977 |
| | IDW | 4.319 | 17.491 | 9.523 | 0.904 | 9.495 |
| | RBF | 7.241 | 12.855 | 9.552 | 0.812 | 8.503 |
| Cd | Sampling point | 0.058 | 0.589 | 0.223 | 0.066 | 29.462 |
| | OK | 0.088 | 0.340 | 0.219 | 0.034 | 15.482 |
| | IDW | 0.094 | 0.489 | 0.224 | 0.024 | 10.703 |
| | RBF | 0.095 | 0.483 | 0.222 | 0.026 | 11.912 |
| Hg | Sampling point | 0.012 | 0.142 | 0.043 | 0.020 | 46.420 |
| | OK | 0.020 | 0.078 | 0.039 | 0.010 | 24.715 |
| | IDW | 0.014 | 0.124 | 0.041 | 0.008 | 19.090 |
| | RBF | 0.019 | 0.104 | 0.041 | 0.007 | 16.828 |
| Mn | Sampling point | 211.000 | 861.000 | 494.798 | 121.866 | 24.629 |
| | OK | 384.795 | 620.448 | 505.262 | 38.888 | 7.697 |
| | IDW | 278.110 | 783.120 | 502.477 | 42.466 | 8.451 |
| | RBF | 362.467 | 601.000 | 500.514 | 36.693 | 7.331 |
| Pb | Sampling point | 11.500 | 33.300 | 24.756 | 3.100 | 12.523 |
| | OK | 21.236 | 28.069 | 24.377 | 1.163 | 4.771 |
| | IDW | 13.710 | 32.727 | 24.513 | 1.271 | 5.187 |
| | RBF | 14.644 | 32.891 | 24.564 | 1.163 | 4.734 |
| Mo | Sampling point | 0.272 | 1.240 | 0.554 | 0.176 | 31.775 |
| | OK | 0.411 | 0.788 | 0.533 | 0.070 | 13.204 |
| | IDW | 0.321 | 1.103 | 0.536 | 0.081 | 15.082 |
| | RBF | 0.319 | 1.085 | 0.534 | 0.084 | 15.795 |

### 3.6. Effects of Different Interpolation Methods on the Assessment Results of Soil Toxic Element Pollution

The data obtained through various interpolation methods were evaluated using the potential ecological risk index method, and the proportion of polluted area was calculated. Concurrently, these results were compared and analyzed alongside those in Table 6. The results are presented in Table 11, and the distribution map of polluted area formed in ArcGIS is shown in Figure 4. All sampling points for As, Mn, Pb, and Mo elements are classified at a light pollution level, and the three interpolation methods do not significantly influence the pollution assessment of these elements.

**Table 11.** Comparison of pollution assessment results of different interpolation methods for toxic elements in soil.

| Element | Method | Light (%) | Moderate (%) | Strong (%) | Very Strong (%) | Extremely Strong (%) |
|---|---|---|---|---|---|---|
| As | Sampling point | 100.000 | 0.000 | 0.000 | 0.000 | 0.000 |
| | OK | 100.000 | 0.000 | 0.000 | 0.000 | 0.000 |
| | IDW | 100.000 | 0.000 | 0.000 | 0.000 | 0.000 |
| | RBF | 100.000 | 0.000 | 0.000 | 0.000 | 0.000 |
| Cd | Sampling point | 21.930 | 75.440 | 2.630 | 0.000 | 0.000 |
| | OK | 9.474 | 90.526 | 0.000 | 0.000 | 0.000 |
| | IDW | 0.827 | 99.138 | 0.035 | 0.000 | 0.000 |
| | RBF | 4.675 | 95.273 | 0.052 | 0.000 | 0.000 |
| Hg | Sampling point | 58.770 | 39.040 | 2.190 | 0.000 | 0.000 |
| | OK | 79.706 | 20.294 | 0.000 | 0.000 | 0.000 |
| | IDW | 77.874 | 22.118 | 0.008 | 0.000 | 0.000 |
| | RBF | 77.056 | 22.942 | 0.002 | 0.000 | 0.000 |
| Mn | Sampling point | 100.000 | 0.000 | 0.000 | 0.000 | 0.000 |
| | OK | 100.000 | 0.000 | 0.000 | 0.000 | 0.000 |
| | IDW | 100.000 | 0.000 | 0.000 | 0.000 | 0.000 |
| | RBF | 100.000 | 0.000 | 0.000 | 0.000 | 0.000 |
| Pb | Sampling point | 100.000 | 0.000 | 0.000 | 0.000 | 0.000 |
| | OK | 100.000 | 0.000 | 0.000 | 0.000 | 0.000 |
| | IDW | 100.000 | 0.000 | 0.000 | 0.000 | 0.000 |
| | RBF | 100.000 | 0.000 | 0.000 | 0.000 | 0.000 |
| Mo | Sampling point | 100.000 | 0.000 | 0.000 | 0.000 | 0.000 |
| | OK | 100.000 | 0.000 | 0.000 | 0.000 | 0.000 |
| | IDW | 100.000 | 0.000 | 0.000 | 0.000 | 0.000 |
| | RBF | 100.000 | 0.000 | 0.000 | 0.000 | 0.000 |

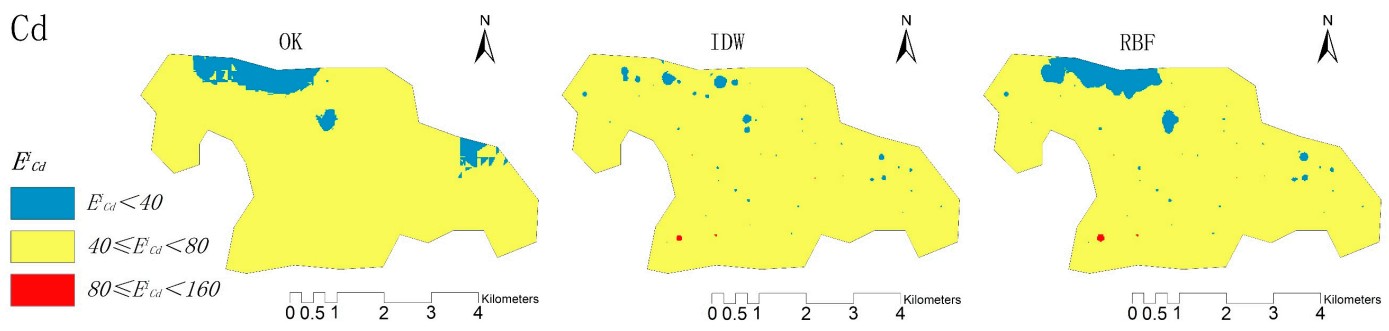

**Figure 4.** *Cont.*

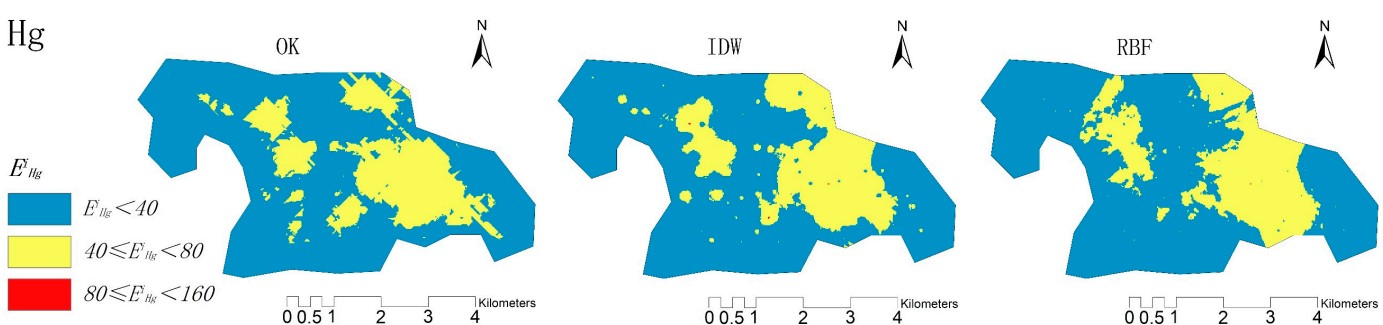

**Figure 4.** Distribution of potential ecological risk pollution areas of toxic elements in soil after interpolation by three interpolation methods.

For Cd, the three interpolation methods all increased the acreage of moderate pollution area ($40 \leq E_{Cd}^i < 80$) and reduced the proportion of light pollution area ($E_{Cd}^i < 40$) and strong pollution area ($80 \leq E_{Cd}^i < 160$). The order of decrement in the proportion of light pollution is IDW (21.103%) > RBF (17.255%) > OK (12.456%). It can be observed in the spatial distribution map that the OK method eliminates the area of strong pollution due to its smoothing effect, which significantly impacts the pollution assessment. For the acreage of light pollution areas, OK is comparable to RBF and the predicted pollution areas interpolated by the two methods is larger than IDW.

For Hg, all three methods increased the proportion of light pollution area ($E_{Hg}^i < 40$) and reduced the proportion of moderate pollution area ($40 \leq E_{Hg}^i < 80$) and strong pollution area ($80 \leq E_{Hg}^i < 160$). The order of amplification for light pollution is OK (20.936%) > IDW (19.104%) > RBF (18.286%), and the order of decrement for moderate pollution was OK (18.746%) > IDW (16.922%) > RBF (16.089%). OK also does not present the prediction of strong pollution area. For the light and moderate pollution area, the predictions of the three methods are similar.

In general, all three methods exhibit the phenomenon of increasing the acreage of the pollution area with larger proportion, and the increases in the three methods are similar. This phenomenon of data concentration and accuracy decline, caused by the decrease in data discreteness, is also a manifestation of the smoothing effect. The greater the increase or decrease, the more pronounced the smoothing effect.

### 3.7. Correlation and Source Analysis of Toxic Elements in Soil

The results of the correlation analysis, including the correlation coefficient matrix and the Pearson correlation coefficient for toxic elements, are presented in Figure 5. The results showed that the correlations between As and Mo (r = 0.66), As and Pb (r = 0.54), Mo and Pb (r = 0.54) are high. The correlations between As and Hg (r = 0.21), As and Mn (r = 0.015), Cd and Hg (r = 0.28), Cd and Mn (r = 0.13), Cd and Pb (r = 0.078), Hg and Mo (r = 0.44), Hg and Pb (r = 0.43) are general. There is a negative correlation between As and Cd (r = −0.25), Cd and Mo (r = −0.23), Hg and Mn (r = −0.16), Mn and Mo (r = −0.15), Mn and Pb (r = −0.23).

The positive matrix factorization (PMF) model was used to analyze the pollution sources of soil toxic elements in the study area. The PMF's input data comprised concentration data and corresponding uncertainty data. The studied elements were classified as "strong" based on their signal-to-noise ratio (S/N). Upon inputting four source factors into the program, the Q value stabilized, indicating the model's applicability [50]. Concurrently, the fitting degree ($R^2$) for As, Cd, Hg, Mn, Pb, and Mo reached 0.862,0.700, 0.924, 0.977, 0.997, and 0.632, respectively, indicating its applicability and reliability of the results [50]. The contribution rates and average contribution rates of each factor are shown in Table 12 and Figure 6. It can be found that the major toxic elements contributed by factor 1 are Cd and Hg, with contribution rates of 45.8% and 71.9%, respectively. The main toxic elements contributed by factor 2 are Mn and As, with contribution rates reaching 49.9% and 37.6%. The major toxic elements contributed by factor 3 are As, Pb, and Mo,

with contribution rates of 50.6%, 40.0%, and 50.5%, respectively. The major toxic element contributed by factor 4 is Cd, with a contribution rate of 54.2%.

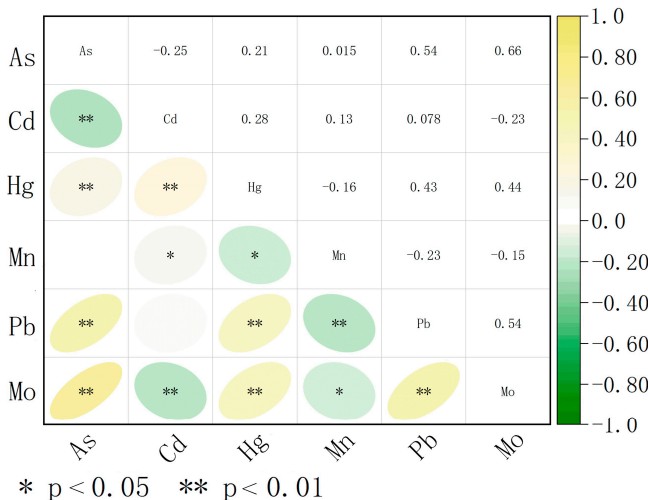

* p < 0.05   ** p < 0.01

**Figure 5.** Correlation analysis of soil toxic elements.

**Table 12.** Contribution rates of soil toxic element pollution factors in the study area.

| Element | Factor Contribution Rate (%) | | | |
|---|---|---|---|---|
| | Factor 1 | Factor 2 | Factor 3 | Factor 4 |
| As | 11.7 | 37.6 | 50.6 | 0.1 |
| Cd | 45.8 | — | — | 54.2 |
| Hg | 71.9 | — | 28.1 | — |
| Mn | 15.7 | 49.9 | — | 34.4 |
| Pb | 18.2 | 13.5 | 40.0 | 28.2 |
| Mo | 21.6 | 27.8 | 50.5 | 0.1 |

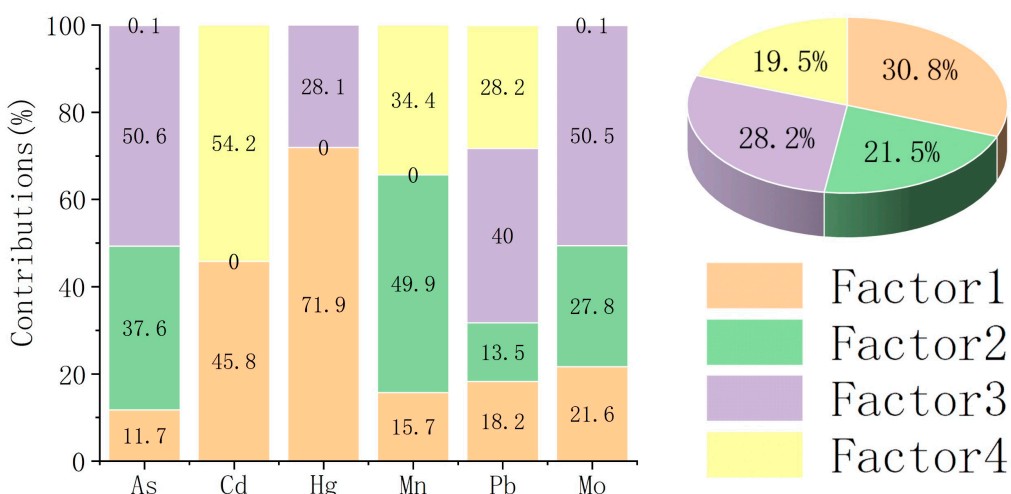

**Figure 6.** Contribution rates and average contribution rates of PMF source apportionment factors of soil toxic elements in the study area.

### 3.8. Spatial Distribution of Hotspots of Toxic Elements in Soil

Based on the source analysis of the soil toxic elements, the contribution rate of each source factor was calculated to further explore the relationship between the main source factor of each toxic element and its spatial distribution. Using the contribution values of the main source factors from the PMF model for the toxic elements of 228 sampling

points, the spatial distribution of hotspots of soil toxic elements was analyzed. The results are presented in Figure 7. It can be observed that the spatial distributions of As, Mo, and Pb hotspots are very similar, suggesting a possible common source for these three elements. The high hotspots (99%, 95%, and 90% confidence intervals) are concentrated in the construction land and cultivated land in the central and northern parts of the study area (Figure 7a,e,f). The high hotspots (99%, 95%, and 90% confidence intervals) of Cd are mainly concentrated in cultivated land and forest land in the eastern and southern parts of the study area (Figure 7b). The high hotspots (99%, 95%, and 90% confidence intervals) of Hg are mainly concentrated in the construction land in the middle of the study area, and a small number of high hotspots are distributed in the northeast and northwest of the study area (Figure 7c). The high hotspots (99%, 95%, and 90% confidence intervals) of Mn are less, mainly distributed in woodland and cultivated land in the northern part of the study area (Figure 7d).

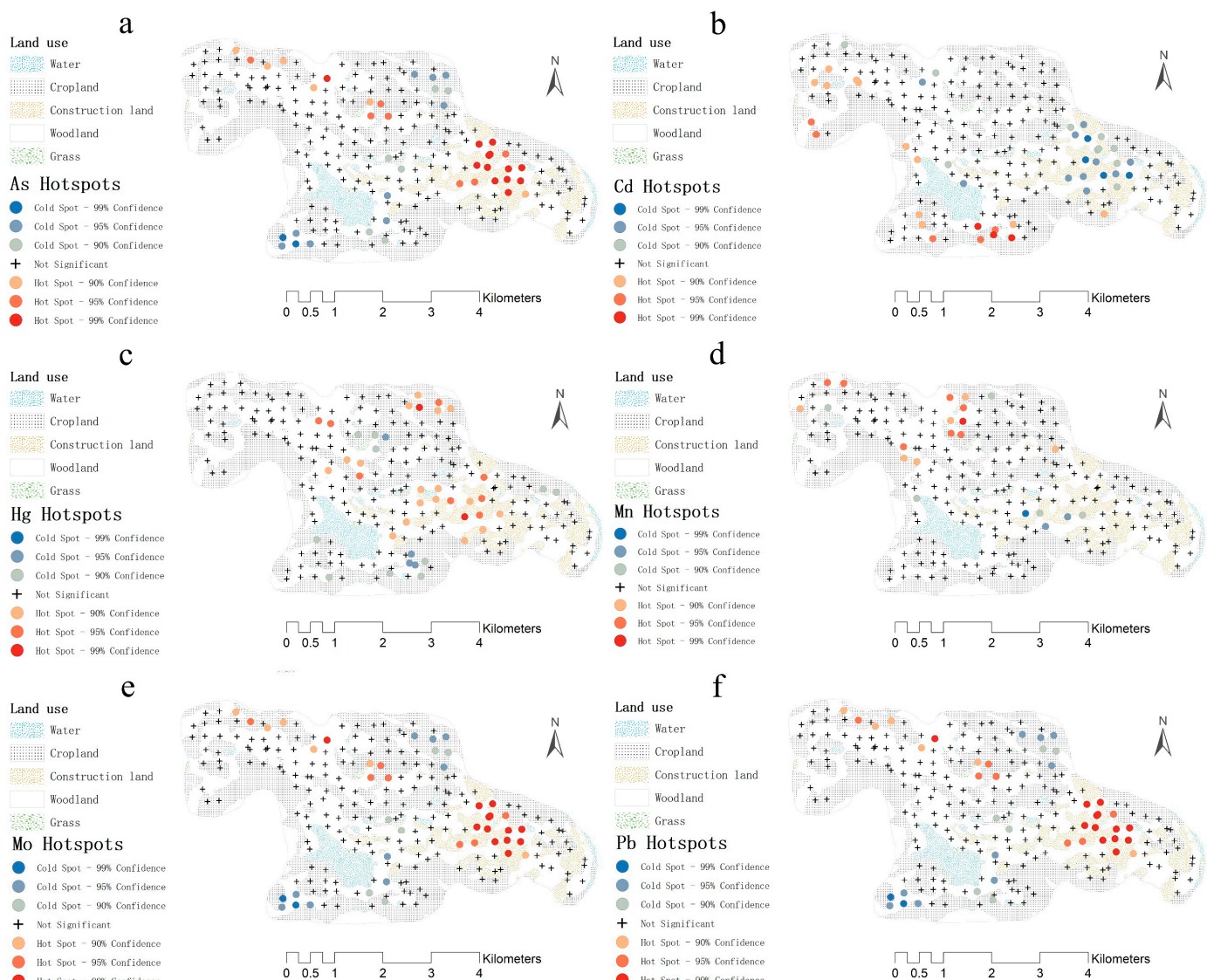

**Figure 7.** (**a**) The spatial distribution of As hotspots under the influence of factor 3; (**b**) the spatial distribution of Cd hotspots under the influence of factor 4; (**c**) the spatial distribution of Hg hotspots under the influence of factor 1; (**d**) the spatial distribution of Mn hotspots under the influence of factor 2; (**e**) the spatial distribution of Mo hotspots under the influence of factor 3; and (**f**) the spatial distribution of Pb hotspots under the influence of factor 3.

## 4. Discussion

### 4.1. Comparative Analysis of Toxic Element Concentration and Pollution Status in Soil

Considering the Chinese and Chengdu soil background values as benchmarks, the soil toxic element pollution in the study area was relatively moderate. Only the mean concentration of Cd simultaneously exceeded the Chinese soil background value by 2.299 times and the Chengdu soil background value by 1.715 times, while the mean concentration of Pb exceeded 7.635% of the Chengdu soil background value. Compared to other parts of China (as shown in Table 13), our study area has a relatively high concentration of As. From the perspective of pollution assessment, the mean $E_r^i$ of As is relatively low and does not reach the level of moderate pollution (Table 3). The mean concentration of Cd in this study area is higher than that of many other study areas, with an $E_r^i$ higher than those recorded in Lanzhou [51], and Shuozhou [52], suggesting a need for long-term local monitoring. Compared to other selected study areas, the mean concentration of Hg in this study area is relatively high. Although the average $E_r^i$ dose not reach the moderate pollution level (Table 3), it still needs long-term monitoring. The average soil concentration and $E_r^i$ of Mn in this study area are similar to those in other study areas [51,53,54], indicating almost no ecological risk (Table 3). The average concentration of Pb in this study area is relatively high, with the exception of the Luoyang study area [55], which records a significantly higher average concentration. However, its average potential ecological risk index is similar to those of other study areas, suggesting a mild pollution level (Table 3). The data on Mo are relatively limited, with this study area recording a low concentration indicative of light pollution (Table 3). In general, the RI in this study area is lower than that in many other study areas, suggesting a light pollution level (Table 3). This implies that the soil toxic element pollution in this study area is not severe.

**Table 13.** Comparison of toxic element concentration and pollution status in the study area of Cangxi and other research reports.

| Item | As | | Cd | | Hg | | Mn | | Pb | | Mo | | RI | Reference |
|---|---|---|---|---|---|---|---|---|---|---|---|---|---|---|
| | Mean (mg/kg$^{-1}$) | $E_r^i$ | Mean (mg/kg$^{-1}$) | $E_r^i$ | Mean (mg/kg$^{-1}$) | $E_r^i$ | Mean (mg/kg$^{-1}$) | $E_r^i$ | Mean (mg/kg$^{-1}$) | $E_r^i$ | Mean (mg/kg$^{-1}$) | $E_r^i$ | | |
| Cangxi, Guangyuan, China | 9.735 | 7.489 | 0.223 | 51.454 | 0.043 | 36.827 | 494.798 | 0.581 | 24.756 | 5.382 | 0.554 | 13.848 | 115.581 | This study |
| Luoyang, China | — | — | 5.870 | 293.270 | — | — | — | — | 155.070 | 4.560 | — | — | 309.310 | Hui et al. [55] |
| The Xingqing Park in Xi'an, China | 5.760 | 5.200 | — | — | — | — | 574.390 | 1.000 | 56.970 | 13.300 | — | — | 179.100 | Guo et al. [53] |
| Lanzhou, Northwestern China | 4.860 | 1.940 | 0.150 | 7.450 | — | — | 534.650 | 8.190 | 16.700 | 0.490 | 0.540 | — | 25.610 | Zeng et al. [51] |
| Songnen-Plain, Northeastern China | 7.160 | — | 0.080 | — | 0.020 | — | 439.150 | — | 24.700 | — | 0.970 | — | 106.000 | Sun et al. [54] |
| Shuozhou, China | 9.269 | 11.880 | 0.117 | 31.270 | 0.030 | 71.280 | — | — | 21.328 | 7.720 | — | — | 124.270 | Yan et al. [52] |
| the Xiaohe River Irrigation Area of the Loess Plateau, China | 13.080 | 12.69 | 0.410 | 138.660 | 0.260 | 698.140 | — | — | 37.210 | 12.680 | — | — | 882.12 | Meng et al. [56] |
| Orchard Soils s in Shaanxi Province, Northwest China | 11.400 | — | 0.290 | — | 0.050 | — | — | — | 23.4 00 | — | — | — | — | Dong et al. [57] |

### 4.2. The Comparison of Spatial Interpolation Methods

Diverse impacts on the spatial dispersion of soil toxic elements are produced by different interpolation methods. As shown in Table 9, with optimal parameters, the interpolation accuracy of OK, IDW, and RBF for each toxic element can be observed. Generally, the *ME*, *RMSE*, and *IP* values of the same toxic element after interpolation using different methods are similar, whereas the accuracy of the same interpolation method can vary greatly with different toxic elements. Overall, the accuracy of OK is relatively high. For the prediction of As, Mn, and Pb, OK has obvious advantages in accuracy, because it produces the smallest *IP*. While the *ME*, *RMSE*, and *IP* values for Cd and Hg obtained using the OK, IDW, and

RBF methods are similar, the IDW method exhibits relatively higher accuracy despite the slight differences. The interpolation of Mo using the RBF method results in an *ME* value closer to 0 and smaller *RMSE* and *IP* values, making it a relatively optimal choice.

The statistics of the interpolation results across the three methods reveal a common trend: the range, standard deviation (SD), and coefficient of variation (CV) decrease, while the mean value changes compared to the original sampling point data. This can be attributed to the inherent smoothing effect of these interpolation methods. The influence of smoothing effect on interpolation results can be evaluated by the variation range of each value. As indicated in Table 10, the OK method has the most pronounced smoothing effect, with IDW and RBF showing similar effects. This may be because both IDW and RBF methods are types of deterministic interpolation and aim to retain the actual value of the sampling point data [18]. The three interpolation methods exhibit distinct spatial distribution effects, as evidenced by the generated spatial distribution map. As shown in Figure 3, the OK method provides the best prediction for spatial distribution of Hg, with a more detailed depiction of varying concentration ranges. The spatial distribution for As, Mn, and Pb predicted by IDW is the best. RBF has the best spatial distribution for the prediction of Cd and Mo. Studies have shown that OK can show a significant smoothing effect in areas with great changes in element content and poor spatial autocorrelation [58]. Therefore, the smoothing effect of OK on the spatial distribution of As, Cd, Mn, Pb, and Mo in this study is stronger than that of the other two methods, while OK can predict the spatial distribution of Hg in detail, which may be because of the high Moran's index and high spatial autocorrelation of As, Cd, Mn, Pb, and Mo (Table 14) [59]. The variation system (46.545%) of Hg sampling points was higher, which means that the content of Hg in this study changed greatly. At the same time, the nugget coefficient ($C_o/(C_o + C)$) was higher (2.580) and Moran's index (0.056) was lower, indicating that the spatial autocorrelation of Hg was poor (Table 14) [39,59].

**Table 14.** Spatial autocorrelation parameters of soil toxic elements.

| Element | Moran's Index | z-Score | p-Value |
|---------|---------------|---------|---------|
| As | 0.148 | 3.669 | 0.000 |
| Cd | 0.076 | 1.935 | 0.053 |
| Hg | 0.056 | 1.456 | 0.145 |
| Mn | 0.065 | 1.658 | 0.097 |
| Pb | 0.221 | 5.414 | 0.000 |
| Mo | 0.240 | 5.865 | 0.000 |

Given that the pollution assessment of most soil toxic elements indicates a light pollution level, this study focuses on Cd and Hg, two elements with a higher pollution degree, to illustrate the impact of OK, IDW, and RBF on the pollution assessment results. The smoothing effect of OK results in the disappearance of strongly polluted areas with the smallest acreage, leading to inaccurate predictions of more severe pollution zones, which negatively impacts local environmental protection and management. Ultimately, RBF has a minimal impact on the pollution assessment of Cd, and similarly, IDW's influence on the assessment of soil Hg pollution is negligible. It is apparent that OK, despite having lower RMSE and IP values, does not predict polluted areas more accurately than the other two methods. This suggests that RMSE and IP, as measures of prediction accuracy, fail to capture the estimation error of local extremes. Furthermore, this study finds that a lower IP value corresponds to a more pronounced smoothing effect. These findings align with the results of a study conducted by Xie et al. (2011) [60].

*4.3. Analysis of Influencing Factors and Sources*

In this study, the PMF model was applied to analyze the sources of toxic element pollution. The analytical findings are presented in Figure 6. The primary sources of

pollution comprise four main factors. The average contribution rates of factors 1, 2, 3, and 4 are 30.8%, 21.5%, 28.2%, and 19.5%, respectively.

According to Table 12, the main toxic elements in factor 1 are Cd (45.8%) and Hg (71.9%). Studies have shown that industrial activities such as mining, smelting, and urban solid waste are the main sources of Cd in soil [61–64]. Hg pollutants in soil in China are mainly derived from man-made industrial activities such as non-ferrous metal smelting, coal combustion, coal mining, and slag activities [65,66]. According to the spatial distribution of Hg hotspots under the influence of factor 1 (Figure 7c), the high hotspot values are mainly concentrated in construction land, which mainly includes rural settlement and industrial and mining land. Simultaneously, considering the significant changes in Hg content and its poor spatial correlation, factor 1 could potentially represent the source from industrial activities.

The main toxic elements in factor 2 are Mn (49.9%) and As (37.6%). However, the Pearson correlation between As and Mn is not high, which suggests As and Mn may not have the same source. The concentration values of Mn in most sampling points are lower than the soil background value of Chengdu. Simultaneously, Mn, being a loading element, is likely to be derived from soil parent material [50,67–69]. Considering the spatial distribution of Mn hotspots influenced by factor 2 (Figure 7d), the high hotspots of Mn are sporadic and far away from the construction land. Additionally, the Moran's index of Mn is relatively low (0.065), which means the spatial autocorrelation is poor. The source may be from the natural environment, thus, factor 2 may represent the source of soil parent material.

The main toxic elements in factor 3 are As (50.6%), Pb (40.0%), and Mo (50.5%). In the correlation analysis, the Pearson correlation coefficient of these three toxic elements is also high, which may indicate that As, Mo, and Pb have a similar pollution level or are released from the same pollution source [70,71]. Additionally, they are mainly affected by the same factor, which also confirms the possibility for same source of As, Pb, and Mo. According to the spatial distribution of As, Pb, and Mo hotspots under the influence of factor 3 (Figure 7a,e,f), the spatial distribution of the hotspots of the three elements is similar, and the high hotspot values are concentrated in the construction land of the study area. Meanwhile, As and Pb pollution in soil may come from coal combustion and atmospheric deposition of industrial by-products [63,72]. Coal combustion in the rural settlement, incorporated into the construction land, may be the primary source of As and Pb in the soil of the study area. Additionally, one of the main sources of soil Pb and Mo pollution is traffic exhaust [73,74]. The high hotspot area of Pb and Mo contains several crucial local traffic roads, suggesting that soil Pb and Mo may result from the deposition of traffic exhaust. In summary, factor 3 may represent the source of atmospheric deposition caused by coal burning and automobile exhaust.

The main toxic element in factor 4 is Cd (54.2%). In addition to industrial activities, Cd is a significant element in pesticides, phosphate fertilizers, and animal manure [63,75,76]. Simultaneously, according to the spatial distribution of Cd hotspots under the influence of factor 4 (Figure 7b), high hotspot values are concentrated in woodland and cultivated land near the construction land. The main source of pollution may include pesticides and fertilizers. Therefore, factor 4 may represent the source of agricultural activity.

## 5. Conclusions

In this study, the toxic element content of soil surface was measured at 228 sampling points in Cangxi County, Guangyuan City. According to the measurement results, three different spatial interpolation methods were used. The results revealed significant errors following the use of the three types of interpolation. Compared with the measured values, the overall interpolation error was significant, but the difference between OK, IDW, and RBF for the same toxic element was relatively minor. These results may be associated with human activities and changes in environmental spatial distribution. Comparing *RMSE*, *ME*, and *IP* of OK, IDW, and RBF, it can be found that OK is more accurate in

interpolation, followed by RBF and IDW. Simultaneously, the three interpolation methods have varying degrees of smoothing effects that influence the prediction of the maximum and minimum values, spatial distribution maps and the pollution assessment. In general, the smoothing effect of OK is more convenient than the other two methods, while IDW and RBF are similar. OK has the most significant impact on the prediction of maximum and minimum values, and the impact of IDW is slightly more pronounced than that of RBF. The three interpolation methods indicate similar spatial distribution characteristics. However, due to the differences in predicted values at the same position, the shape and acreage of different content intervals vary. Regarding the impact on pollution assessment, all three methods show a trend of increasing the acreage of the pollution area proportionately, with each method causing a different rate of increase. Based on the accuracy of interpolation, the interpolation results, the spatial distribution maps and the impact on pollution assessment, the most suitable method for each toxic element in soil can be determined. The optimal interpolation method for As, Hg, and Mn is IDW; for Cd and Mo, it is RBF; and for Pb, it is OK. Therefore, it is necessary to select the most suitable interpolation parameters and method according to the environmental factors such as soil characteristics and human factors in the study area.

In analyzing the source of soil toxic elements in the study area, we used correlation analysis, PMF model, and hotspot analysis. The analysis findings show that the main source is human activities, further subdivided into industrial activities (30.8%), potentially a significant source of soil Cd and Hg pollution, atmospheric deposition caused by coal burning and traffic exhaust (21.5%) which may be a major source of soil As, Pb, and Mo pollution and agricultural activities (19.5%) which may be a major source of soil Cd pollution. However, the soil parent material, as a natural source, which may be a major source of soil Mn pollution, only contributes 28.2% of the average contribution rate. In the future, due to the strong impact of human activities, there may be a continuous increase in the concentration of soil toxic elements in the study area. The goal of this study is to provide a scientific foundation for local soil remediation, environmental management, and conservation, aiming to enhance local sustainability.

**Author Contributions:** Conceptualization, J.Z. and J.P.; methodology, J.P. and X.S.; software, X.S.; validation, J.Z., J.P. and W.C.; formal analysis, Z.F.; investigation, X.C. and Y.L.; resources, X.C. and Y.L.; data curation, J.Z. and X.S.; writing—original draft preparation, J.Z., J.P. and X.C.; writing—review and editing, W.C.; visualization, Y.M.; supervision, W.C.; project administration, W.C.; funding acquisition, W.C. All authors have read and agreed to the published version of the manuscript.

**Funding:** This paper thanks the support of Sichuan Science and Technology Support Program "Soil Heavy metal Pollution and Ecological Risk Prediction in Hilly Area of Central Sichuan" (Approval No.: 2014SZ0068); "Construction of Ecological Security Pattern in Chengdu–Chongqing Double City Economic Circle" (Approval No.: 22RKX0490).

**Institutional Review Board Statement:** Not applicable.

**Informed Consent Statement:** Not applicable.

**Data Availability Statement:** Data are contained within the article.

**Acknowledgments:** The authors are grateful to the anonymous reviewers for their constructive comments and suggestions that helped to improve the quality of this paper.

**Conflicts of Interest:** The authors declare no conflicts of interest. The funders had no role in the writing of the manuscript; or in the decision to publish the results.

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
