# Peer review of "Comparative Study on Different Interpolation Methods and Source Analysis of Soil Toxic Element Pollution in Cangxi County, Guangyuan City, China"

_sustainability, doi:10.3390/su16093545_

Round 1
Reviewer 1 Report
Comments and Suggestions for Authors
Include in the conclusions the possible direct sources of the contaminants studied, in order to provide clearer information.
Author Response
Cover letter
Jiajun Zhang
Chengdu University of Technology
Chengdu, China
March, 2024
Dear Reviewer,
Thank you for giving us the opportunity to submit a revised draft of the manuscript “Comparative study on different interpolation methods and source analysis of soil toxic element pollution in Cangxi County, Guangyuan City, China (original title: Comparative study on different interpolation methods and source analysis of soil heavy metal pollution in Cangxi County, Guangyuan City, China)” for publication in sustainability. We appreciate the time and effort that you and other reviewers dedicated to providing feedback on our manuscript and are grateful for the insightful comments on and valuable improvements to our paper.
We have incorporated most of the suggestions made by the reviewers. Those changes are highlighted in the manuscript. The yellow highlighted texts are revisions according to the reviewer's comments, the green underlined texts are revisions in terms of language and expression, and the red texts are "toxic element", terms such as "heavy metal" and "metal element", after unified. Please see attachment for a point-by-point response to the reviewers’ comments and concerns.
Thank you again for your consideration of this manuscript.
Sincerely,
Jiajun Zhang

Reviewer 2 Report
Comments and Suggestions for Authors
In this manuscript, the authors did a lot of heavy metal testing and analysis work. The scientific problems of this study are clear, the logic of text is good and the analytical methods are reliable. However, there are still some problems that need to be optimized. So, I recommend to you that this manuscript can be accepted after minor modification. The following are the flaws of this manuscript and revision suggestions:
1. As different spatial interpolation methods, the methods of OK (ordinary kriging), IDW (inverse distance weighting), and RBF (radial basis function) are applicable to different situations.When the author does the spatial interpolation analysis, he can choose the method with the least error. The comparative analysis between methods has no universal significance. So I suggest that the intermediate process of the author choosing the most suitable interpolation method can be deleted in the paper (or shown with supplementary materials).
2. It is suggested that the inspection limits of each heavy metal should be list in the Table 1.
3. It is suggested that the National standard values of heavy metal pollution should be listed in the Table 4 and then do a comparative analysis.
Comments on the Quality of English LanguageMinor editing of English language required
Author Response

(The authors gave the same response as above.)

Reviewer 3 Report
Comments and Suggestions for Authors
Thank you for the opportunity to review this paper.
Heavy metal pollution is mostly due to various human activities and major worldwide problem. The identification of effective methods for heavy metal content analysis, interpretation of results through statistical analysis, and obtaining a high-precision spatial distribution map of heavy metals in contaminated soils by spatial interpolation are addressed and presented in this manuscript by the authors.
In the Materials and Methods chapter, the study methods proposed by the authors are presented, methods that were chosen correctly and in accordance with the purpose of the work.
The obtained results are presented in detail; the statistical analysis of the obtained data is well documented. The presentation of the research results was made through numerous tables and figures that clearly explain the distribution of heavy metals in the study area. The authors selected the most appropriate interpolation method for each metal in the study area.
The Discussion chapter reports its own results to existing data in the literature.
The manuscript is large and complex, it is well scientifically documented.
The conclusions established by the authors are clear and complex, the conducted study provides a scientific basis for ensuring soil remediation, environmental conservation and management in order to increase economic sustainability.
I recommend that this paper be accepted and published in this journal.
Author Response
Cover letter
Jiajun Zhang
Chengdu University of Technology
Chengdu, China
March, 2024
Dear Reviewer,
Thank you for giving us the opportunity to submit a revised draft of the manuscript “Comparative study on different interpolation methods and source analysis of soil toxic element pollution in Cangxi County, Guangyuan City, China (original title: Comparative study on different interpolation methods and source analysis of soil heavy metal pollution in Cangxi County, Guangyuan City, China)” for publication in sustainability. We appreciate the time and effort that you and other reviewers dedicated to providing feedback on our manuscript and are grateful for the insightful comments on and valuable improvements to our paper.
We have incorporated most of the suggestions made by the reviewers. Those changes are highlighted in the manuscript. The yellow highlighted texts are revisions according to the reviewer's comments, the green underlined texts are revisions in terms of language and expression, and the red texts are "toxic element", terms such as "heavy metal" and "metal element", after unified.
Thank you again for your consideration of this manuscript.
Sincerely,
Jiajun Zhang

Reviewer 4 Report
Comments and Suggestions for Authors
Review comments
The paper by Zhang et al. represents a qualitative scientific work but have few mistakes and unanswered questions. I have few objections. In “Materials and methods” chapter authors need to put information’s about collection of samples according to geochemical survey. The authors use different methods for determination of metals. What about comparison of different methods? Why did authors not describe determination of pH in method part of article?
In discussion part of the article, you mentioned Mn as factor, what about As? Try to use same term for metal. In your text you have: metal element, heavy metal? At the end, I am wondering about references. This topic is actual in other parts of the world not only in China. You have more than 80% of China literature. I encourage authors to revise their references and put some from other parts of the world. Therefore, I recommend acceptation of the paper with major revision.
My specific comments are as follows:
Abstract
L 15: Try to find better word then “to offer”.
L 21: Delete “Than”.
L 22: Change “It turns out” with “It can be concluded”.
L 25: You need to explain PMF.
Introduction
L35: Delete second “sustainability”.
L37-39: This sentence need revision.
L45-47: This sentence need revision.
L56-57: Please put some references about widely used methods.
Materials and methods
L103: You can put numbers for latitudes in brackets.
L105-106: Please use WRB for classification of sampled soils.
L115: Please explain collection of samples according to geochemical survey?
L119: Soil column? I suppose that you collect your samples with some kind of shovel and in disturbed conditions?
L121-122: You do not need weight of samples in article.
L128: “Different methods were used to determine the samples.” This is incorrect. To determine what? Concentration of metals? Revise sentence.
L129: You use different methods for determination of metals. How you can do comparison?
L136: Explain SPSS?
L161-162: This sentence need revision.
L174-175: Put reference for soil background value.
L179: Put reference for table.
Results
L203: Where you described pH measuring method?
L204-205: This sentence is for discussion part of manuscript.
L205-210: This sentence is not result. This part of text belongs to Materials and methods.
L219: From light to moderate?
L236-245: This part of text belongs to Materials and methods.
L260-263: This part of text belongs to Materials and methods.
L284-292: This part of text needs revision.
L307: Change “and” with “while”.
L309-311: This part of text needs revision.
L400-401: This sentence is for discussion part of manuscript.
L413-414: What about As?
Discussion
L450-451: This part of text needs revision.
L456-459: Why you compare your results from soils with sediments or sludge?
L479-480: This part of text needs revision.
L489-491: This part of text needs revision.
L496-497: This part of text needs revision.
L522-523: “The sources of pollution are mainly four factors.” This needs revision.
L535: What about As?
L535: Try to use same term for metal. In your text you have: metal element, heavy metal? What is it?
L545: Change “the three elements” with “they”.
L546: Change “of” with “for”.
Conclusions
L565: Try to use same term for metal. In your text you have: metal element, heavy metal? What is it?
L568: Change “kinds” with “types”.
L576: Change “obvious” with “convenient”.
L587-589: What about soil characteristics?
L590: Change “this paper” with “we used”.
L597-599: This part of text needs revision.
Figures:
Fig. 1. Put higher font for legend.
Fig. 2. Put higher font for axes.
Fig. 3. Put higher fonts for legend and scales.
Fig. 4. Put higher font for scale.
Fig. 5. Put higher font for axes.
Fig. 6. Put higher fonts for legend and scales.
Tables:
Table 1. Explain abbreviations.
Comments on the Quality of English Language
The paper by Zhang et al. represents a qualitative scientific work but have mistakes in English language. Therefore, I recommend to use one of softwares for English editing or nature English speaker.
Author Response

(The authors gave the same response as above.)

Reviewer 5 Report
Comments and Suggestions for Authors
1) The authors should explain why different methods of analysis were used to determine the elemental composition (Table 1) and whether it is permissible to make such an assessment.
2) What specific analytical equipment was used to determine the elemental composition of the samples?
3) It is worth paying attention to a study on similar topics [https://doi.org/10.1016/j.chemosphere.2010.09.053] and provide a corresponding comparison with the results of the current article (or analyze it in the Introduction).
Author Response

(The authors gave the same response as above.)

Reviewer 6 Report
Comments and Suggestions for Authors
The paper is well prepared and contains valuable analysis. I suggest to accept it while taking into account the following minor remarks:
Introduction
Lines 85-88. A description of study area, in the form which food is grown, is not relevant for this section
Material and methods
Table 1. Units for MDL must be indicated
pH is not mentioned in this section though the results of its measuring are presented in the following sections.
Results
Line 207. The statement "The higher CV is, the stronger the influence of human activities is" must be explained more since it is not evident.
Discussion
Table 13. Units for concentrations must be indicated
Author Response

(The authors gave the same response as above.)

Round 2
Reviewer 4 Report
Comments and Suggestions for Authors
Dear Authors, I accept manuscript in this form. Kind regards, Reviewer
Author Response
Cover letter
Jiajun Zhang
Chengdu University of Technology
Chengdu, China
April, 2024
Dear Reviewer,
Thank you agian for giving us the opportunity to submit a revised draft of the manuscript “Comparative study on different interpolation methods and source analysis of soil toxic elements pollution in Cangxi County, Guangyuan City, China (original title: Comparative study on different interpolation methods and source analysis of soil heavy metal pollution in Cangxi County, Guangyuan City, China)” for publication in sustainability. We appreciate the time and effort that you and other reviewers dedicated to providing feedback on our manuscript and are grateful for the insightful comments on and valuable improvements to our paper. We have gained valuable insights from our engagements with you.
Thank you again for your consideration of this manuscript.
Sincerely,
Jiajun Zhang
